# Unique Interactions of the Small Translocases of the Mitochondrial Inner Membrane (Tims) in *Trypanosoma brucei*

**DOI:** 10.3390/ijms25031415

**Published:** 2024-01-24

**Authors:** Linda S. Quiñones, Fidel Soto Gonzalez, Chauncey Darden, Muhammad Khan, Anuj Tripathi, Joseph T. Smith, Jamaine Davis, Smita Misra, Minu Chaudhuri

**Affiliations:** 1Department of Microbiology, Immunology, and Physiology, School of Medicine, Meharry Medical College, Nashville, TN 37208, USA; lquinones17@email.mmc.edu (L.S.Q.); fidel.gonzalez@nih.gov (F.S.G.); mkhan@mmc.edu (M.K.); anuj1.tripathi@yahoo.com (A.T.); 2Department of Biochemistry, Cancer Biology, Neuroscience, and Pharmacology, School of Medicine, Meharry Medical College, Nashville, TN 37208, USA; cdarden17@email.mmc.edu (C.D.); jdavis@mmc.edu (J.D.); 3Department of Microbiology and Immunology, Jacobs School of Medicine and Biomedical Sciences, University at Buffalo, Buffalo, NY 14203, USA; jsmith85@buffalo.edu; 4Department of Biomedical Science, School of Graduate Studies, Meharry Medical College, Nashville, TN 37208, USA; smisra@mmc.edu

**Keywords:** mitochondria, translocase, small Tims, protein interactions, *Trypanosoma brucei*

## Abstract

The infectious agent for African trypanosomiasis, *Trypanosoma brucei*, possesses a unique and essential translocase of the mitochondrial inner membrane, known as the TbTIM17 complex. TbTim17 associates with six small TbTims (TbTim9, TbTim10, TbTim11, TbTim12, TbTim13, and TbTim8/13). However, the interaction patterns of these smaller TbTims with each other and TbTim17 are not clear. Through yeast two-hybrid (Y2H) and co-immunoprecipitation analyses, we demonstrate that all six small TbTims interact with each other. Stronger interactions were found among TbTim8/13, TbTim9, and TbTim10. However, TbTim10 shows weaker associations with TbTim13, which has a stronger connection with TbTim17. Each of the small TbTims also interacts strongly with the C-terminal region of TbTim17. RNAi studies indicated that among all small TbTims, TbTim13 is most crucial for maintaining the steady-state levels of the TbTIM17 complex. Further analysis of the small TbTim complexes by size exclusion chromatography revealed that each small TbTim, except for TbTim13, is present in ~70 kDa complexes, possibly existing in heterohexameric forms. In contrast, TbTim13 is primarily present in the larger complex (>800 kDa) and co-fractionates with TbTim17. Altogether, our results demonstrate that, relative to other eukaryotes, the architecture and function of the small TbTim complexes are specific to *T. brucei*.

## 1. Introduction

African trypanosomiasis, a fatal disease impacting humans and livestock in sub-Saharan African countries, is caused by the parasitic protozoan *Trypanosoma brucei* [1,2,3]. In eukaryotes, mitochondria play essential roles in many cellular functions critical for cell viability [4,5,6]. This is especially true for trypanosomatids, which only possess one mitochondrion [7,8]. Mitochondria house hundreds of proteins, with the majority (98%) of these proteins encoded in the nucleus; thus, they need to be imported into mitochondria after synthesis in the cytosol, emphasizing the importance of the biological process of mitochondrial protein import [9]. The machinery for mitochondrial protein import consists of multi-protein complexes, extensively studied in fungi and mammals [10,11]. We and others have identified a non-canonical protein import apparatus in *T. brucei*, belonging to a group of early divergent eukaryotes [12,13]. Therefore, this unique mitochondrial complex warrants thorough investigation, not only for identifying novel chemotherapy targets but also for elucidating the evolutionary perspective of this essential cellular process.

In yeast, the mitochondrial protein import machinery consists of three main complexes: translocase of the outer membrane (TOM) and two translocases of the inner membrane (TIM), namely TIM22 and TIM23 [10]. The TIM23 complex imports matrix-targeted proteins with the assistance of a presequence translocase-associated motor (PAM) complex [14,15]. A few mitochondrial inner membrane (MIM) proteins imported via the TIM23 complex have a sorting signal and are laterally sorted into the MIM [10]. The TIM22 complex, on the other hand, imports multi-spanning MIM proteins [16,17]. Small Tim complexes, including Tim8, Tim9, Tim10, Tim12, and Tim13, chaperone these hydrophobic proteins from the TOM to the TIM22 complex [18,19,20,21]. These small Tims, ranging from 8 to 12 kDa, are soluble proteins in the intermembrane space (IMS) and are also associated with the TIM22 complex (Figure 1A). *Saccharomyces cerevisiae* has five small Tims: Tim8, Tim9, Tim10, Tim12, and Tim13. These small Tims possess the highly conserved twin CX_3_C motifs, with cysteine residues forming two intramolecular disulfide bonds. Yeast small Tims form three heterohexameric complexes of ~70kDa, (Tim9)_3_-(Tim10)_3_, (Tim9)_3_-(Tim10)_2_-(Tim12), and (Tim8)_3_-(Tim13)_3_ [22,23]. Tim12 associates with the TIM22 translocase, aiding in docking the Tim9-Tim10 complex carrying cargo proteins to the TIM22 complex for further translocation [10,22,23]. Humans have six small Tims: Tim8a, Tim8b, Tim9, Tim10a, Tim10b, and Tim13 [24]. They also form similar heterohexameric complexes, with human Tim10b acting similarly to yeast Tim12 [24]. In yeast, Tim9, Tim10, and Tim12 are essential, whereas Tim8 and Tim13 are apparently dispensable [18,19,20]. However, it has been shown that Tim8-Tim13 plays a role in the assembly of Tim23 into the MIM in yeast [25,26], as well as the voltage-dependent anion channel (VDAC) into the MOM in *Neurospora crassa* [27]. Meanwhile, in humans, mutations of Tim8 lead to the neurodegenerative disorder known as the human deafness dystonia syndrome [28]. Additionally, Tim8a, Tim8b, and Tim13 also play roles in the assembly of complex IV in human mitochondria [29].

The crystal structures of yeast and human Tim9-Tim10 complexes and the human Tim8-Tim13 complex revealed that, in each complex, two monomers are alternately arranged, forming a donut-shaped structure [18,30,31]. The core region of this structure consists of the central loop and a pair of disulfide bonds that form the flat face along the molecular axis, and the N- and C-termini that emanate from the opposite face as tentacle-like structures. Both hydrophobic and electrostatic interactions hold these monomers together [31]. Particularly, salt bridges formed between highly conserved glutamic acid and lysine residues are found to be critical for the stability of this complex [23].

In contrast to fungi and mammals, *T. brucei* utilizes two major protein complexes for mitochondrial protein import (Figure 1B) [12,13]: the archaic TOM complex (ATOM) in the MOM and the TbTIM17 complex in the MIM [32,33,34,35]. Besides TbTim17, the TbTIM17 complex in *T. brucei* possesses several unique components. [33,34,35]. Additionally, *T. brucei* has six small TbTims (TbTim9, TbTim10, TbTim11, TbTim12, TbTim13, and TbTim8/13) [33,36,37,38,39]. TbTim9, TbTim10, and TbTim8/13 were initially identified through a homology search in the *T. brucei* genome database using Hidden Markov prediction models, with yeast and human small Tims as the queries [36]. Distinctively, Tim8/13 contains features present in both Tim8 and Tim13 in other eukaryotes. Subsequently, other small TbTims (TbTim11, TbTim12, and TbTim13) were identified with SILAC-coimmunoprecipitation analysis of TbTim17 and found to be unique to trypanosomes [33]. Except for TbTim12, each of these small TbTims possesses conserved twin CX_3_C motifs. Although TbTim12 has one Cysteine (C) in each motif, alpha-fold prediction (www.alphafold.ebi.ac.uk, accessed on 22 July 2021) revealed similar secondary and tertiary structures with small Tims in yeast and humans. Orthologous proteins have been found in other trypanosomatids; however, there is limited information about the small Tims in these species [40]. The small TbTims are soluble IMS proteins crucial for *T. brucei* cell growth and associated with the TbTIM17 complex [37,38]. In *T. brucei*, the TbTIM17 complex imports both N-terminal- and internal targeting signal-containing nuclear-encoded proteins destined for the mitochondrial matrix or the MIM, respectively, as well as tRNAs needed for mitochondrial translation (Figure 1B) [33,34,39,41,42]. Previous studies suggest that small TbTims are present in multiple complexes, including two smaller complexes (~150 kDa and ~70 kDa) and in a larger complex (>800 kDa) similar in size to the TbTIM17 complex [37,38]. However, it is unclear whether these proteins randomly interact to form these complexes or if there is a specific pattern of interactions and functions of the small TbTims within *T. brucei*. In this study, we systematically analyzed the interaction pattern and complexes formed by the small TbTims and compared the effect of RNAi of each individual small TbTim on the integrity of the TbTIM17 complex. Altogether, we found that TbTim9, TbTim8/13, and TbTim10 interact strongly and may form the core small TbTim complex. Except for TbTim13, small TbTims formed 70 kDa complexes, whereas TbTim13 only associates with the larger complex (>800 kDa) along with TbTim17. Moreover, TbTim13 is the most essential among all small TbTims for TbTIM17 complex biogenesis/stability. Therefore, despite overall structural homology with the small Tims in other eukaryotes, the interaction pattern, and the architecture of the small TbTim complexes differ in *T. brucei*.

## 2. Results

### 2.1. Small TbTims Interact with Each Other with Stronger Interactions among TbTim9, TbTim10, and TbTim8/13

*T. brucei* possesses six small TbTims (TbTim9, TbTim10, TbTim8/13, TbTim11, TbTim12, and TbTim13) and these are known to associate with the TbTIM17 complex [37,38]. To determine the interaction pattern of the small TbTims, we employed yeast two-hybrid (Y2H) analysis as detailed in the Materials and Methods. To distinguish between stronger and weaker interactions, we utilized 3-amino-1,2,4-triazole (AT), a histidine synthase inhibitor at different concentrations (0 mM, 2 mM, 3.5 mM, and 5 mM). The results revealed that yeast cells co-transformed with different pairs of small TbTim clones grew well at 0 mM AT (Figure 2A–E, Plate I), suggesting direct interactions among all small TbTims. However, growth varies with increasing AT concentration (Plates II, III, and IV) (Figure 2A–E, Table 1). Yeast cells co-transformed with the plasmid pairs: TbTim9 and TbTim8/13 (Figure 2A, pair 7), TbTim10 and TbTim8/13 (Figure 2A, pair 8), TbTim8/13 and TbTim8/13 (Figure 2A, pair 5), and TbTim9 and TbTim13 (Figure 2D, pair 6) grew up to 5.0 mM AT, indicating strong interactions. In contrast, the TbTim9 and TbTim10 (Figure 2A, pair 6) and TbTim13 and TbTim13 (Figure 2E, pair 4) pairs exhibited growth up to 3.5 mM AT, indicating moderate interactions. The TbTim11 and TbTim9 (Figure 2B, pair 3), TbTim11 and TbTim10 (Figure 2B, pair 4), TbTim11 and TbTim11 (Figure 2B, pair 5), TbTim11 and TbTim12 (Figure 2B, pair 6), TbTim11 and TbTim13 (Figure 2C, pair 3), TbTim8/13 and TbTim11 (Figure 2C, pair 4), TbTim10 and TbTim12 (Figure 2C, pair 6), TbTim12 and TbTim12 (Figure 2D, pair 3), TbTim13 and TbTim12 (Figure 2D, pair 4), TbTim8/13 and TbTim12 (Figure 2D, pair 5), TbTim10 and TbTim13 (Figure 2E, pair 3), and TbTim8/13 and TbTim13 (Figure 2E, pair 5) pairs showed growth up to 2.0 mM AT, indicating weaker interactions. Lastly, the TbTim9 and TbTim9 (Figure 2A, pair 3), TbTim10 and TbTim10 (Figure 2A, pair 4), and TbTim9 and TbTim12 (Figure 2C, pair 5) transfectants grew only at 0 mM AT, showing very weak interactions. Notably, TbTim8/13 demonstrated strong interactions with TbTim9 and TbTim10 (Figure 2A–F, Table 1), consistent with our previous report (38). We also found moderate interactions between TbTim13 and TbTim9 (Figure 2D). The overall interaction patterns among small TbTims are illustrated in a schematic (see Figure 2F).

To confirm the expression of AD and BD fusion proteins with small TbTims in yeast, we grew co-transformants in SD (-leu, -tryp) liquid medium for 48 h. Subsequently, cells were harvested and total cellular proteins were analyzed via SDS-PAGE and immunoblot analysis using anti-Myc and anti-HA antibodies. We observed that each of these small TbTims fusion proteins were expressed at the expected size in the corresponding co-transfected yeast cells (Figure 2G–I). Duplicate Coomassie-stained gels served as our loading control. The expression levels for most of the fusion proteins were comparable in all transfectants, suggesting that differential interaction patterns are likely due to varying affinity among these pairs.

### 2.2. Co-Immunoprecipitation Analyses from T. brucei Mitochondrial Extract Revealed That TbTim10 Exhibits a Stronger Association with TbTim9 and TbTim8/13, While Showing a Weaker Association with TbTim13

Our Y2H data further indicates that small TbTims display differential interaction patterns with each other. To verify if this holds true in *T. brucei* mitochondria, we performed co-immunoprecipitation analysis of the small TbTims from the *T. brucei* mitochondrial lysate. Given the limited availability of suitable antibodies for the small TbTims, we expressed C-terminally tagged small TbTims in pairs within *T. brucei*. Next, we measured the growth kinetics of these cells following doxycycline induction to determine if the overexpression of small TbTims has any impact on the cells. The expression of TbTim10-Myc with TbTim9-HA, TbTim10-HA, or TbTim8/13-HA showed no difference in cell growth (Appendix A). However, co-expression of TbTim10-Myc with TbTim12-HA or TbTim13 HA inhibited cell growth, particularly after day 2 post-induction. Therefore, we limited the induction period to 2 days for all cell lines in subsequent experiments. We then examined the mitochondrial localization of the tagged small TbTims through immunoblot analysis of sub-cellular fractions. Myc- and HA-tagged small TbTims were localized in the mitochondria with comparable expression levels, except for reduced TbTim10-Myc levels when co-expressed with TbTim12-HA or TbTim13-HA (Appendix A).

Mitochondrial fractions isolated at day 2 post-induction were solubilized with 1% digitonin. Soluble extracts were subjected to immunoprecipitation using either anti-Myc- or anti-HA-coupled agarose beads and subsequently analyzed via immunoblot analysis using Myc and HA antibodies. Positive and negative controls included TbTim17 and VDAC, respectively. The results indicated that Myc antibodies precipitated TbTim10-Myc from all samples, as anticipated (Figure 3A, upper panel). Additionally, these antibodies also pulled down TbTim9-HA and TbTim8/13-HA at approximately ~35% and 45% of the input, respectively. However, they precipitated TbTim10-HA and TbTim12-HA at around ~25% and ~30% of the input, respectively (Figure 4A,B). Contrastingly, TbTim13-HA did not co-precipitate with TbTim10-Myc using anti-Myc antibodies (Figure 3A,B), suggesting a weaker association between TbTim10 and TbTim13; however, this could be attributed to low levels of the TbTim10-Myc in this cell line. Additionally, we observed that TbTim10-Myc co-precipitated a significant fraction of TbTim17 from all samples, except when TbTim13-HA was co-expressed with TbTim10-Myc (Figure 3A). As anticipated, VDAC did not co-precipitated with TbTim10-Myc. Similarly, no protein bands were detected by the HA antibody from samples that only expressed TbTim10-Myc.

In a similar manner, when immunoprecipitating HA-tagged small TbTims from the corresponding samples, we observed that TbTim9-HA and TbTim8/13-HA co-precipitated a fraction of TbTim10-Myc (45% of the input). However, TbTim10-HA, TbTim12-HA, and TbTim13-HA precipitated less than 10% of TbTim10-Myc (Figure 3A, lower panel, and Figure 3C). This further indicates that the interactions of TbTim10 with TbTim9 and TbTim10 with TbTim8/13 are stronger compared to other small TbTims (Figure 3A,C). This also correlates with our Y2H results. The HA antibodies did not precipitate any protein from the mitochondrial extract that only had TbTim10-Myc. Similarly, TbTim17 was co-precipitated from all samples alike. As expected, VDAC was not precipitated by HA antibody from any samples. Due to some technical difficulties, we failed to generate a TbTim11-HA-expressing *T. brucei* cell line; therefore, we could not speculate on the association of this protein with other small TbTims.

### 2.3. TbTim17 C-Terminal Domain Directly Interacts with All of the Small TbTims

While it has been shown that the small TbTims are associated with the TbTIM17 complex [37,38], it remains unclear if they directly interact with TbTim17 and, if so, which regions of TbTim17 are involved. Similar to Tim17/22/23 family proteins, TbTim17 possesses four predicted transmembrane domains (TMs) (TMPred, a transmembrane prediction server) [43], with the N-terminal (1–30 AAs) and C-terminal (140–152 AAs) hydrophilic regions, along with loop-2 (93–102 AAs), the connecting region between TM2 and TM3, exposed in the IMS (Figure 4A) [44]. As small TbTims are localized in the IMS [37,38], we selected these regions of TbTim17 for testing their interactions with small TbTims through Y2H analysis, as described in the Materials and Methods. Yeast cells co-transformed with the N-terminal region of TbTim17, and individual small TbTim grew well up to 2.0 mM AT but reduced at 3.5 mM AT and showed significant hindrance at 5.0 mM AT (Figure 4B), indicating a weaker interaction. Yeast cells transformed with the TbTim17 loop-2 clone and individual small TbTim clones showed minimal growth even at 0 mM AT (Figure 4C). Only a mild growth was observed for the yeast cells transformed with the TbTim17 loop-2 and TbTim11 plasmids. The C-terminal region of TbTim17 strongly interacted with all small TbTims, with co-transformants growing well up to 5 mM AT (Figure 4D, Table 2). Direct interactions between the full length TbTim17 and small TbTims-BD were examined (Figure 4E). However, except for TbTim17 and TbTim12 (Figure 4E, pair 6) and TbTim17 and TbTim8/13 (Figure 4E, pair 8), which grew at 0 mM AT only (Figure 4E, Table 2), none of these small TbTims showed interactions with the full-length TbTim17, possibly due to improper folding of this hydrophobic protein.

Immunoblot analysis confirmed the expression of TbTim17 and its fragments in yeast (Figure 4F). Full-length TbTim17 expressed as TbTim17-HA-AD, the N- and C-terminal hydrophilic regions, and loop2 of TbTim17 expressed as -Myc-BD fusion proteins were observed within expected sizes. TbTim9-Myc-BD and TbTim9-HA-AD were also expressed in the corresponding cell lines. We noticed a duplicate band with closer molecular sizes for TbTim9-HA that was not seen in Figure 2H,I. It is likely that the duplicate bands were generated during preparation of the samples. The expression of other small TbTims in yeast is shown in Figure 2. Notably, the expression levels of the TbTim17-loop2-Myc-BD and TbTim17-AD were relatively lower than the N- and C-terminal fragments, but the terminal fragments were expressed at comparable levels. These results suggest that the C-terminal region of TbTim17 is likely responsible for interacting with small TbTims.

### 2.4. Among All Small TbTims, TbTim13 Is Most Essential for TbTim17 Complex Stability

To assess if knockdown of any of these small TbTims has a stronger effect than others, we compared the levels of the TbTim17 and TbTIM17 complex in mitochondria isolated after each of these six small TbTims’ knockdowns in *T. brucei*, individually. The effects of the knockdown of TbTim9, TbTim10, and TbTim8/13 on cell growth and transcript levels were previously published [38]. Here, we show that TbTim11, TbTim12, and TbTim13 RNAi reduced the levels of the corresponding target transcript about 90%, 60%, and 90%, respectively, by 48 h post-induction (Appendix A). We observed a significant growth inhibition of *T. brucei* cells due to the knockdown of TbTim11, TbTim12, or TbTim13, specifically at day 4 post-induction of RNAi and after (Appendix A). Immunoblot analysis of the mitochondrial protein using the anti-TbTim17 antibody revealed that, at day 2 post-induction, TbTim17 levels were moderately reduced in all small TbTim RNAi samples; however, maximum reduction (~40%) was observed for TbTim13 RNAi (Figure 5A,B). This trend was continued at day 4 post-induction, with TbTim17 levels reduced below 20% of the parental control (Figure 5A,B) in TbTim13 knockdown mitochondria. At this time point, TbTim10 and TbTim8/13 knockdown also reduced TbTim17 levels to 40%, and knockdown of TbTim9, TbTim11, and TbTim12 reduced the levels of TbTim17 to 60–70%. RNAi for individual small TbTims affected VDAC levels similarly. In comparison to the control, mHsp70 levels increased due to knockdown of small TbTims, which could be a stress response phenomenon, since the small TbTims are essential for the *T. brucei* procyclic form; however, this needs to be further investigated. Small TbTims’ knockdown reduced VDAC levels, as we reported earlier, suggesting a role for the small TbTims in VDAC biogenesis (Figure 5A). Tubulin was used as a loading control (Figure 5A).

Next, we compared the levels of the TbTIM17 complex in small TbTim RNAi mitochondria using Blue-Native (BN)-PAGE followed by immunoblot analysis with the TbTim17 antibody. Similar to the steady-state levels of the TbTim17 protein, the TbTIM17 complex levels were significantly reduced due to TbTim13 knockdown at day 2 and more so at day 4 post-induction (Figure 5C). TbTim10 and TbTim8/13 knockdown also significantly reduced TbTIM17 complex levels, relatively more compared to TbTim9, TbTim11, and TbTim12 knockdown at day 4 post-induction (Figure 5C). These results suggest that TbTim13 is the most crucial for the stability or biogenesis of the TbTIM17 complex. TbTim10 and TbTim8/13 also play a significant role in this process, and other small TbTims may be secondarily involved in maintaining the TbTIM17 complex’s integrity. Coomassie-stained gels were used as our loading control. Overall, by comparing the effects of RNAi, we observed that the six small TbTims have a differential effect on TbTim17 complex integrity, suggesting a specific role for the individual small TbTims in this process.

### 2.5. Each of the Small TbTims Is a Component of the Smaller 70 kDa Small TbTim Complexes, with the Exception of TbTim13, Which Is Present in a Larger Complex (>800 kDa) and Co-Fractionates with TbTim17

We utilized size exclusion chromatography (SEC) to analyze the complexes formed by the small TbTims. *T. brucei* cells expressing various small TbTims with Myc and HA tags (TbTim10-Myc and TbTim9-HA, TbTim10-Myc and TbTim10-HA, TbTim10-Myc and TbTim8/13-HA, TbTim10-Myc and TbTim12-HA, and TbTim10-Myc and TbTim13-HA) were used for these analyses. After solubilizing isolated mitochondria with 1% digitonin, soluble supernatants were fractionated using a gel-filtration column (SEC 650, 10 × 300). Each fraction (1–18) was analyzed via SDS-PAGE and immunoblot analysis using Myc-, HA-, and TbTim17 antibodies. Results from fractions 1–13 (equivalent to vol 37–50 mL) are shown (Figure 6A,C,E,G,I,K). Other fractions (14–18) were negative to the antibodies tested. TbTim10-Myc was enriched in fractions 3 and 6–7, while TbTim17 was enriched in fraction 2, in mitochondrial extracts from TbTim10-Myc (Figure 6A,B). Column calibration with known molecular weight marker proteins revealed sizes of ~66 kDa and 1964 kDa for the protein complexes in fractions 6 and 2, respectively (Appendix A). Therefore, TbTim10-Myc was observed in a smaller complex (~70 kDa) and a larger complex, similar in size to the TbTIM17 complex. Co-expression of TbTim8/13-HA with TbTim10-Myc showed that both TbTim10 and TbTim8/13 were co-eluted in fractions 6 and 7 (Figure 6C,D). Our co-immunoprecipitation results support this observation, demonstrating co-precipitation of the two small TbTims when pulled down by either Myc or HA antibodies (Figure 3). Therefore, these two small TbTims are likely present in the same (~70 kDa) complex. A part of TbTim8/13-HA was also detected in fraction 2, along with TbTim17 (Figure 6C,D). Similarly, analysis of the mitochondrial extract from TbTim10-Myc- and TbTim9-HA-expressed *T. brucei* revealed co-elution of TbTim9-HA and TbTim10-Myc in fractions 6 and 7, with a portion eluting alongside TbTim17 in fractions 2 and 3 (Figure 6E,F). Co-expression of TbTim12-HA with TbTim10-Myc resulted in a larger fraction of TbTim12-HA enriched in fractions 6 and 7 (Figure 6G,H). However, a significant part of TbTim10-Myc was found in complexes with intermediate sizes. This suggests that, in the presence of abundant TbTim12-HA, the 70 kDa complex is formed with TbTim12, and excess TbTim10-Myc associates with the other intermediate complexes. (Figure 6G,H). Co-expression of TbTim10-Myc with TbTim10-HA and analysis of the mitochondrial protein complexes via SEC 650 showed TbTim10-HA behaving similarly to TbTim12-HA, primarily eluting with fractions 6 and 7. However, TbTim10-Myc levels were reduced in these fractions, being instead found in the intermediate complexes (Figure 6I,J). This implies that, due to constitutive expression of TbTim10-HA, most of this protein is found in the 70 kDa complex, while a part of the TbTim10-Myc, induced with doxycycline, is in the intermediate complexes. In contrast, TbTim13-HA was mainly enriched in fraction 2 with TbTim17 and not in fractions 6 or 7, indicating that TbTim13-HA does not form a smaller (70 kDa) complex (Figure 6K,L). As mentioned earlier, in TbTim10-Myc + TbTim13-HA cells, levels of TbTim10-Myc were significantly reduced, resulting in a reduced elution in fraction 2 (Figure 6K,L). These findings suggest that overexpressing TbTim13 diminishes the assembly and stability of TbTim10. Immunoblot analysis of the column fractions using mHsp70 antibodies revealed mHsp70 elution across fractions 2–8 (Figure 6M), consistent with its known association with multiple mitochondrial complexes [45,46]. Notably, mHsp70 enrichment in fractions 6 and 7, likely representing its monomeric form, was consistent across all samples. In contrast, VDAC primarily eluted in fractions 2 and 3, indicating its multimeric form, as expected (Figure 6M), and this pattern was consistent across all samples. Combining these results along with our co-precipitation and RNAi data, we propose that: (1) TbTim13 is not present in a smaller (70 kDa) complex but rather associates only with the larger TbTIM17 complex, and it is crucial for TbTIM17 complex assembly/stability. (2) TbTim10 and TbTim13 may compete for binding to TbTim17, with TbTim13 having stronger binding, leading to a dissociation of TbTim10 from the complex and reducing its stability. (3) TbTim9, TbTim10, TbTim8/13, and TbTim12 each forms the smaller (70 kDa) complex. (4) Simultaneous expression of TbTim10-Myc (inducible) and TbTim10-HA (constitutive) results in TbTim10-HA predominantly in the 70 kDa complex, while excess TbTim10-Myc is mainly present in the intermediate complexes. This aligns with our co-immunoprecipitation results, showing limited co-precipitation of TbTim10-Myc and TbTim10-HA. These results support a model for the small TbTim complexes (Figure 7), where TbTim9, TbTim10, TbTim8/13, and TbTim12 form multiple heterohexameric complexes. TbTim10 and TbTim12 may exist in distinct 70 kDa complexes, while TbTim13 consistently associates with the TbTIM17 complex. The small TbTims heterohexameric complexes could exchange subunits and dynamically associate with the larger TbTIM17 complex for protein translocation.

## 3. Discussion

The TbTIM17 complex is unique in *T. brucei* in multiple aspects. It consists of several trypanosome-specific proteins, and it is involved in the import of both the N-terminal MTS-containing and the internal targeting signal-containing hydrophobic MIM proteins, as well as tRNAs [34,39,41,42]. While it is speculated that TbTim17 carries out these complex functions by dynamically associating with different components, the complete picture remains unclear. *T. brucei* possesses six essential small TbTims, discovered as parts of this complex; however, information about their interaction pattern among themselves and with TbTim17 was limited. Through Y2H analysis and co-immunoprecipitation studies, we demonstrated that TbTim8/13 exhibits stronger interactions with TbTim9 and TbTim10 compared to others, while TbTim13 shows a stronger association with TbTim17. This observation was also supported by analysis of the small TbTim complexes using SEC, which revealed that TbTim9, TbTim8/13, TbTim10, and TbTim12 can form 70 kDa complexes, which could be similar to the heterohexameric complexes of the small Tims found in yeast and mammals [22,23], while TbTim13 remains associated with the larger TbTIM17 complex, crucial for its integrity. Furthermore, we discovered that the C-terminal domain of TbTim17 likely serves as a point of contact with the small TbTims, while the N-terminal domain weakly interacts with a few small TbTims. Hence, TbTim13 may act as ScTim12 or HsTim10b, forming a stable part of the larger translocase complex, while the other small TbTims form the characteristic 70 kDa complex in different combinations, dynamically associating with the larger TbTIM17 complex. These studies increased our understanding about the single TIM complex in *T. brucei*.

After comparing the effect of knocking down of each of the small TbTims in parallel, we observed that the depletion of TbTim13 has a more drastic effect on the levels of the TbTIM17 complex, as well as TbTim17 steady-state protein levels, compared to the knockdown of other small TbTims. TbTim10 and TbTim8/13 also showed a similar reduction in TbTIM17 complex, but it required a longer RNAi induction period. Wenger et al. investigated TbTim11, TbTim12, and TbTim13, noting that TbTim13 is crucial for maintaining TbTIM17 complex levels, though not TbTim17 protein levels [37]. However, our experiments revealed a significant loss of TbTim17 protein due to TbTim13 RNAi. This discrepancy might be attributed to variations in the extent of knockdown or different laboratory strains of *T. brucei*. The notable decrease in the steady-state levels of TbTim17 in our experiment is possibly because the unassembled TbTim17 was degraded in the absence of TbTim13.

Analysis of the small TbTim complexes through SEC analysis revealed that TbTim10 eluted alongside TbTim9 or TbTim8/13 in fractions 6–7, suggesting the potential formation of heterohexameric complexes. The stoichiometry of the subunits in the 70 kDa complex needs to be further determined; however, we can assume that, like their counterparts in yeast and human, it could be the (TbTim10)_3_-(TbTim9)_3_, (TbTim10)_3_-(TbTim8/13)_3_, or (TbTim8/13)_3_-(TbTim9)_3_ complexes. The possibility that all three proteins coexist in the same complexes, such as (TbTim8/13)_3_-(TbTim9)_2_-TbTim10 or (TbTim8/13)_2_-(TbTim9)_3_-TbTim10, cannot be ruled out. Notably, TbTim10-HA showed poor co-precipitation with TbTim10-Myc, with TbTim10-HA saturating the 70 KDa complex, preventing assembly of excess TbTim10-Myc in smaller complexes after induction. Similarly, co-expressed TbTim12-HA and TbTim10-Myc were present in the 70 kDa complex, but excess TbTim10-Myc was also found in the larger intermediate complexes, suggesting that TbTim12 may share common constituents but not be directly associated with TbTim10-Myc in the 70 kDa complex. TbTim10 and TbTim13 did not co-precipitate from mitochondrial extract, but TbTim13 strongly co-precipitated with TbTim17. Additionally, TbTim13 co-eluted with TbTim17 in fractions 2–3 by SEC, not in fractions 6–7, leading to the conclusion that TbTim13 associates with the TbTIM17 complex and is not present in the 70 kDa complex.

The human TIM22 complex structure has recently been resolved by Cryo-EM at a 3.7 Å resolution [47]. This structure revealed that the N-terminal helices of HsTim22 interact with 2 heterohexameric small Tim complexes, Tim9_3_-Tim10_3_ and Tim9_2_-Tim10a_3_-Tim10b. Other components of the HsTIM22 complex, such as acylglycerol kinase and Tim29, also interact with the small Tim complexes [47]. The stoichiometric ratios of the components in the TbTIM17 complex are currently unclear. Given the abundance of TbTim17 in the TbTIM complex, it suggests the presence of more than one copy of TbTim17 per complex. In Y2H analysis, we found that the C-terminus of TbTim17 interacts more strongly with all the small TbTims than the N-terminus, indicating that the C-terminal end likely anchors the small TbTim complexes. The lack of interaction with full-length TbTim17 is probably due to its nature as a hydrophobic membrane protein, leading to improper folding when expressed in the yeast cytosol or upon entering the yeast nucleus. The lower interaction of the TbTim17-loop2 region with small TbTims could be partially attributed to lower expression levels of this fragment. TbTim11 showed weaker interactions with the loop2 region of TbTim17. In our previous studies, we found that TbTim54, a peripherally associated MIM protein, interacts with TbTim11 [39], indicating that TbTim11 could make the connection between TbTim17 and TbTim54. However, future studies by purification of these complexes and Cryo-EM analysis will uncover more details about the structure of this unique translocase complex in *T. brucei*.

Our studies have significantly advanced our understanding of the interaction among small TbTims and their connection with TbTim17, shedding light on the formation of various complexes. Analysis of similarities and differences in the structure and function of small TbTims and TbTIM17 complexes with their functional counterparts in other systems is crucial for gaining mechanistic and evolutionary insights into this conserved cellular process in eukaryotes. Notably, it has been shown that redox-regulated small Tims’ biogenesis can be targeted by small molecule inhibitors [48,49]. Some human small Tims are involved in functions beyond mitochondrial protein import, such as cytochrome oxidase biogenesis, and mutations in Tim8a are linked with the hereditary disorder, Mohr–Tranebjaerg syndrome [28]. Therefore, exploring the interactions and functions of the small Tims in different systems will increase our knowledge and aid in the development of more suitable inhibitors.

## 4. Materials and Methods

### 4.1. Cell Maintenance, Growth Medium, and Cell Growth Analysis

The procyclic form of the *T. brucei* 427 doubly resistant cell line (29–13) expressing a tetracycline repressor gene and a T7 RNA polymerase was grown in SDM-79 medium supplemented with 10% fetal bovine serum, G418 (15 µg/mL), and hygromycin (50 µg/mL) [50]. To measure growth, cells were seeded at a cell density of 3 × 10^6^ cells/mL in fresh medium containing the appropriate antibiotics. Cell numbers were counted at different time points (0–8 days) using a Neubauer hemocytometer. The log of cumulative cell numbers was plotted versus time (in days) of incubation.

### 4.2. Generation of Plasmid Constructs and T. brucei Transgenic Cell Lines

*T. brucei* cell line that expresses TbTim10-Myc upon induction with doxycycline was developed previously [38]. For generation of double-tagged small TbTim cell lines (TbTim10-Myc/TbTim9-HA, TbTim10-Myc/TbTim10-HA, TbTim10-Myc/TbTim12-HA, TbTim10-Myc/TbTim13-HA, and TbTim10-Myc/TbTim8/13-HA), the open reading frames (ORFs) of TbTim9, TbTim10, TbTim12, TbTim13, TbTim8/13 were PCR-amplified using *T. brucei* 427 genomic DNA as the template and the corresponding sequence-specific primers (Appendix A). The forward and reverse primers were designed with addition of restriction sites for *HindIII* and *XbaI* at the 5′ ends, respectively. The PCR products were cloned into the modified pHD1344 vector, where the coding sequence was inserted within the *HindIII* and *XbaI* restriction sites. Generated constructs were linearized by *NotI* digestion and transfected into *T. brucei* TbTim10-Myc cell line as described [38]. Transfected cells were selected with puromycin (1.0 µg/mL). The TbTim9 RNAi, TbTim10 RNAi, and TbTim8/13 RNAi cells were developed previously [38]. The constructs for TbTim11 RNAi, TbTim12 RNAi, and TbTim13 RNAi were generated by PCR amplification of the ORFs of TbTim11, TbTim12, and TbTim13, using *T. brucei* genomic DNA as the template and sequence-specific primers (Appendix A). The forward and reverse primers were designed with addition of restriction sites for *HindIII* and *BamHI* at the 5′ ends, respectively. The PCR products were cloned into tetracycline-inducible p2T7Ti-177 RNAi vector between the *HindIII* and *BamHI* restriction sites [51] Plasmid DNA was linearized by *NotI* digestion and transfected into *T. brucei* 29–13 cells. Transfected cells were selected with puromycin (1.0 µg/mL).

### 4.3. RNA Isolation and Quantitative RT-PCR Analysis

RNA was isolated from *T. brucei* cells using RNeasy miniprep isolation kit (Qiagen, Germantown, MD, USA) and digested with amplification grade DNase (1 U/µL) for 1 h before first-strand cDNA synthesis, which was performed using an iScript cDNA synthesis kit (Bio-Rad, Hercules, CA, USA). The resulting cDNA was amplified using specific primers designed from the 3′-untranslated region sequence of the target genes for small TbTims (Appendix A) to detect endogenous TbTim11, TbTim12, and TbTim13 mRNA levels but not the double-stranded RNA generated from the p2T7^TI^-177 RNAi construct. Primers for amplification of tubulin cDNA were generated from the ORF of the tubulin gene (Appendix A).

### 4.4. Yeast Two-Hybrid Analysis

The ORFs of TbTim11, TbTim12, and TbTim13 were subcloned into yeast expression vectors pGADT7 and pGBKT7 (Takara, San Jose, CA, USA) to generate the bait and prey plasmids [52]. TbTim9, TbTim10, and TbTim8/13 were cloned previously in the pGADT7 and pGBKT7 vectors [38]. The coding regions of TbTim17 N-terminal (1–30 aa), loop2 (140–152 AAs), and C-terminal (93–107 AAs), as well as the ORF of the TbTim17 full-length protein, were subcloned into yeast expression vectors pGADT7 and pGBKT7. Approximately 2 µg of each of the bait and prey plasmids in different combination pairs was co-transformed into the *Saccharomyces cerevisiae* Y2H Gold strain (Takara) using the lithium acetate method [38]. Co-transformed yeast cells were plated on SD medium lacking leucine (-leu) and *tryptophan (-trp)* and allowed to grow for 3 days at 30 °C. Yeast clones that grew were then plated on SD –leu/–trp/–his medium that was lacking leucine, tryptophan, and histidine to select for protein–protein interactions. SD –leu/–trp/–his plates were also supplemented with 2.0, 3.5, and 5.0 mM AT, which inhibits Y2H yeast cell growth due to leaky expression of the *HIS3* gene [53,54], to limit the occurrence of false positives. Inoculated plates were allowed to grow at 30 °C for 3 to 5 days. To confirm positive readouts, this process was repeated at least three times with individual clones.

### 4.5. Sub-Cellular Fractionation and Crude Mitochondria Isolation

Fractionation of *T. brucei* procyclic form cells was performed as described [38]. Briefly, 2 × 10^8^ cells were pelleted and re-suspended in 500 μL of SMEP buffer (250 mM sucrose, 20 mM MOPS/KOH, pH 7.4, 2 mM EDTA, 1 mM PMSF) containing 0.03% digitonin and incubated on ice for 5 min. The cell suspension was then centrifuged for 5 min at 6800× *g* at 4 °C. The resultant pellet was considered as the crude mitochondrial fraction, and the supernatant contained soluble cytosolic proteins. Mitochondria were isolated from the parasite after lysis by nitrogen cavitation in isotonic buffer [38]. The isolated mitochondria were stored at a protein concentration of 10 mg/mL in SME buffer containing 50% glycerol at −70 °C. Before use, mitochondria were washed twice with nine volumes of SME buffer to remove glycerol [38].

### 4.6. SDS-PAGE and Immunoblot Analysis

Proteins from whole-cell lysates or cytosolic or mitochondrial extracts were separated on a 15% SDS polyacrylamide gel, transferred to a nitrocellulose membrane, and immunodecorated with polyclonal antibodies for TbTim17 (Tb927.11.13920), TbAAC (Tb927.10.14820), VDAC (Tb927.2.2510), TbPP5 (Tb927.10.13670) (see in Ref. [34]), mtHsp70 (Tb927.6.3740) [55], and *T. brucei β*-tubulin (Tb927.1.2350) [56]. Anti-Myc and anti-HA polyclonal antibodies were purchased from commercial sources (Abcam, Waltham, MA, USA and Thermo-Fisher, Waltham, MA, USA, respectively). Blots were developed with appropriate secondary antibodies and an enhanced chemiluminescence kit (Thermo-Fisher).

### 4.7. BN-PAGE Analysis

Mitochondrial proteins (200 µg) were solubilized in 100 µL of ice-cold 1× native buffer (Thermo-Fisher) containing 1% digitonin. The solubilized mitochondrial proteins were clarified by centrifugation at 100,000× *g* for 30 min at 4 °C. The supernatants were mixed with G250 sample additive (Thermo-Fisher, Waltham, MA, USA) and were electrophoresed on a precast (4% to 16%) bis-Tris polyacrylamide gel (Thermo-Fisher, Waltham, MA, USA), according to the manufacturer’s protocol. Protein complexes were detected via immunoblot analysis. Molecular size marker proteins apoferritin dimer (886 kDa) and apoferritin monomer (443 kDa), amylase (200 kDa), alcohol dehydrogenase (150 kDa), and bovine serum albumin (66 kDa) were electrophoresed on the same gel and visualized by Coomassie staining. Proteins were transferred to a nitrocellulose membrane for Western blot analysis.

### 4.8. Co-Immunoprecipitation Assay

Mitochondria (600 μg) were solubilized in 300 µL of 1× cold native buffer (50 mM [Tris pH 7.2], 50 mM NaCl, 10% [wt/vol] glycerol, 1 mM PMSF, 1% digitonin) and incubated on ice for 1 h. The solubilized mitochondria were centrifuged at 100,000× *g* for 30 min. An aliquot (50 μL) of the supernatant was mixed with 50 µL of 2× Laemmli sample buffer and served as the input sample. The remaining supernatant (~250 µL) was mixed with 25 µL of anti-Myc-or anti-HA-conjugated agarose bead slurry (Sigma-Aldrich) and allowed to incubate for 12 h at 4 °C with constant gentle inversion. The beads were separated by centrifugation at 1000× *g* for 1 min. An aliquot (50 μL) of the supernatant was mixed with 50 µL of 2× Laemmli sample buffer and served as the unbound fraction. The beads were then washed three times in wash buffer (50 mM Tris [pH 7.2], 50 mM NaCl, 10% [wt/vol] glycerol, 1 mM PMSF, 0.1% digitonin) to remove nonspecifically bound proteins. The washed beads with bound proteins were resuspended in 50 µL of 1× Laemmli sample buffer and served as the bound fraction. The samples were then separated on a 15% SDS polyacrylamide gel and transferred to a nitrocellulose membrane. Blots were developed with appropriate secondary antibodies and an enhanced chemiluminescence kit (Thermo-Fisher, Waltham, MA, USA).

### 4.9. Size Exclusion Chromatography

Mitochondrial proteins (4 mg) were solubilized in 1.25 mL of 1X Native buffer (20 mM Tris, pH 7.0, 50 mM NaCl, 1 mM EDTA, 10% glycerol, 1 mM PMSF, and 1% digitonin). The solubilized proteins were loaded on a ENrich^TM^ SEC 650 10 × 300 column (Bio-Rad, Hercules, CA, USA) and run on an NGC system (BioRad). The proteins were eluted with the same buffer, except the digitonin concentration was reduced to 0.1%. After discarding the void volume (6.0 mL), 18 fractions (1.0 mL each) were collected. Proteins in each fraction were analyzed via SDS-PAGE and immunoblotting using antibodies for Myc, HA, TbTim17, VDAC, and mHsp70. Gel filtration molecular weight markers were run separately on the column under same conditions and were detected in the eluted fractions via SDS-PAGE and CB-staining (Appendix A). The sizes of the small TbTims complexes and TbTim17 complexes were calculated from a standard curve, in which the log10 values of molecular sizes of marker proteins were plotted against the elution volumes.

## Figures and Tables

**Figure 1 ijms-25-01415-f001:**
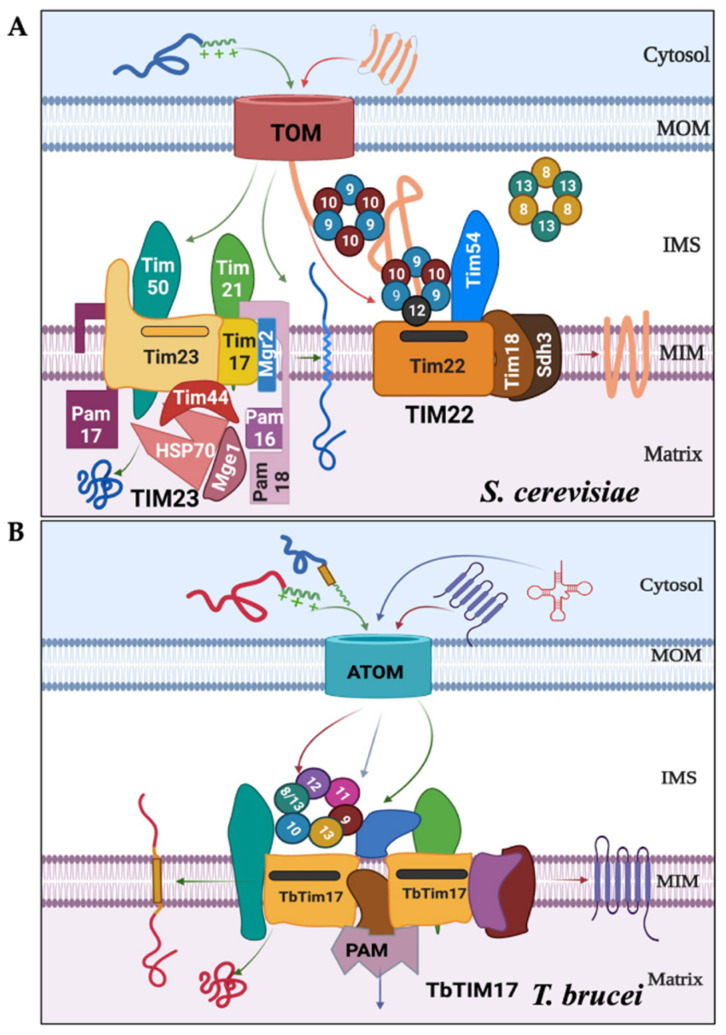
The major protein complexes involved in mitochondrial protein import differ between *S. cerevisiae* and *T. brucei*. (**A**) In fungi and mammals, after crossing the TOM complex, the N-terminal MTS-containing proteins are selected by the TIM23 complex and imported into the matrix with assistance from the presequence translocase-associated motor (PAM) complex. The internal signal-containing proteins are translocated via the TIM22 complex, with assistance from the small Tim complexes (Tim9–Tim10 and Tim8–Tim13) localized in the inter membrane space (IMS). (**B**) In *T. brucei*, both types of signal-containing proteins are transported via ATOM and then through the TbTIM17 complex consisting of TbTim17 and other trypanosome-specific Tims. The TbTIM17 complex consists of unique TbTims and the small TbTims: TbTim9, TbTim10, TbTim8/13, TbTim11, TbTim12, and TbTim13. The tentative import pathways for proteins with N-terminal (green arrows), internal targeting signals (red arrows), and tRNAs (blue arrows) are shown.

**Figure 2 ijms-25-01415-f002:**
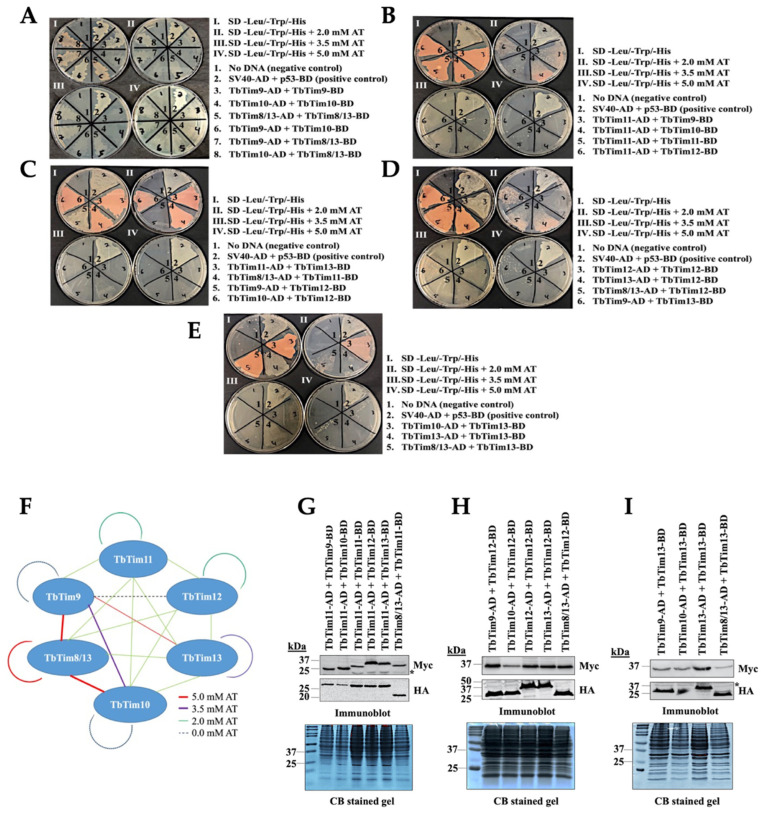
Intermolecular interactions of the small TbTims via Y2H analysis. Six small TbTims were individually cloned in the pGADT7 (Gal4 activation domain, AD) and pGBKT7 (Gal4 DNA-binding domain, BD) plasmids and used as the bait and prey in all possible combinations to investigate their interaction pattern. (**A**–**E**) Yeast strain co-transformed with the bait and the prey plasmids were grown in (I) medium lacking leucine, tryptophan, and histidine (–leu/–trp/–his), (II) –leu/–trp/–his medium containing 2.0 mM 3-amino-1,2,4-triazole (AT), (III) –leu/–trp/–his medium containing 3.5 mM AT, and (IV) –leu/–trp/–his medium containing 5.0 mM AT. Yeast cells co-transformed with no DNA were used as negative controls. Yeast cells co-transformed with SV40-AD and p53-BD were used as the positive control. The plates shown are representatives of three independent experiments. (**F**) A schematic of the small TbTim interaction pattern based on the Y2H analysis results. Red lines represent cell growth observed even at 5.0 mM AT, purple lines represent cell growth observed in the presence of up to 2.0 mM AT, green lines represent cell growth observed at 2.0 mM, and dashed blue lines represent cell growth at 0 mM AT. (**G**–**I**) Expression of the small TbTims-AD and -BD fusion proteins in yeast. Yeast cells co-transformed with a pair of small TbTims-containing plasmids, as indicated, were grown in –Leu/-Trp broth and subjected to protein extraction. Total cellular proteins were analyzed via SDS-PAGE and immunoblot analysis using anti-Myc and anti-HA antibodies to detect BD and AD fusion proteins tagged with the Myc and HA epitopes, respectively. Non-specific bands are indicated by *. Corresponding Coomassie-stained gels are shown for loading control.

**Figure 3 ijms-25-01415-f003:**
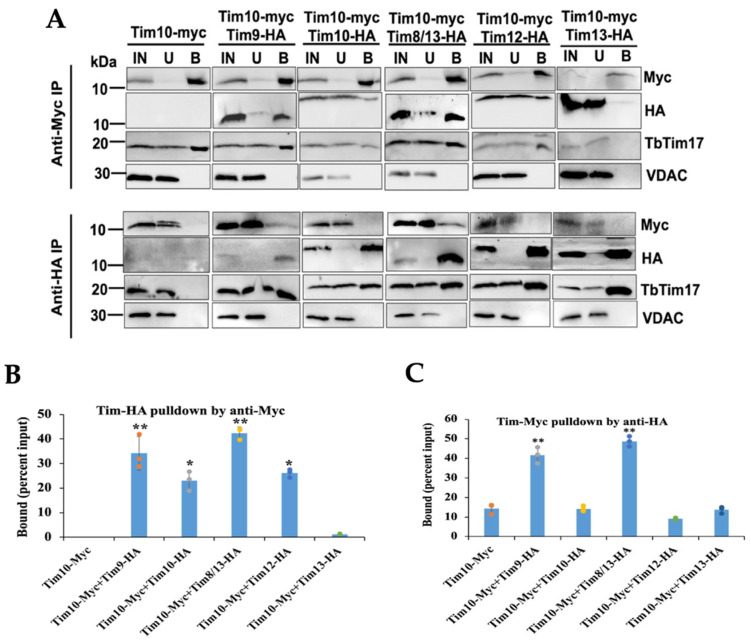
Co-immunoprecipitation analysis of the small TbTims from *T. brucei* mitochondrial extract. (**A**) The digitonin-solubilized mitochondrial extracts from double-tagged small TbTims cells were used for immunoprecipitation using agarose beads coupled with the anti-Myc or anti-HA antibody. Proteins in the input (IN), unbound (U), and bound (**B**) fractions were analyzed on SDS-PAGE, followed by immunoblot analysis using Myc and HA antibodies. Twenty percent (vol/vol) of the IN and U, and 25% (vol/vol) of the B were loaded in the respective lanes. Blots were probed, also with TbTim17 and VDAC antibodies. Samples were run in more than one gel; therefore, results for each cell type are presented in individual panels. (**B**,**C**) Densitometry analysis of the immunoblots using ImageJ software (version 1.53a, National Institute of Health, Baltimore, MD, USA). The quantitated bound fractions (Myc and HA) were calculated as percentage of total input and plotted for each sample. Standard errors were calculated from three independent experiments. Significance values are calculated from *t* test and are indicated by asterisks as follows; ** *p* < 0.01; * *p* < 0.05.

**Figure 4 ijms-25-01415-f004:**
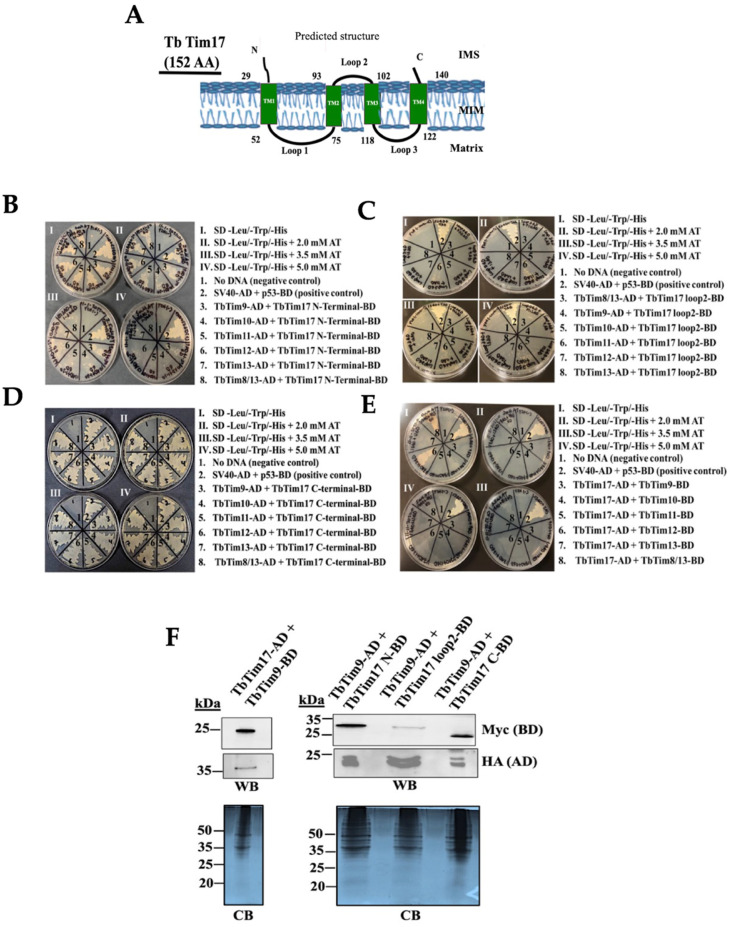
Y2H analysis of the small TbTims’ interactions with TbTim17. (**A**) A schematic of TbTim17 protein. The predicted transmembrane domains (TM1-TM4) and the membrane topology are shown. Numbers indicate the amino acid residues on the protein. (**B**–**D**) The N-terminal (1–30 AAs), loop 2 (93–102 AAs), and the C-terminal (140–152 AAs) of TbTim17 were sub-cloned in pGBKT7 to use as the bait. The small TbTims cloned in the pGADT7 were used as the prey to investigate interactions between each of the TbTim17 fragments with individual small TbTims. (**E**) The full-length (FL) TbTim17 cloned in the pGADT7 and the small TbTims cloned in pGBKT7 were also used in parallel for Y2H analysis. Yeast strain (Gold) co-transformed with the bait and prey plasmids were grown in (I) –leu/–trp/–his, (II) –leu/–trp/–his medium containing 2.0 mM AT, (III) –leu/–trp/–his medium containing 3.5 mM AT, and (IV) –leu/–trp/–his medium containing 5.0 mM AT. Yeast cells co-transformed with no DNA were used as negative controls. Yeast cells co-transformed with SV40-AD and p53-BD were used as the positive control. The plates shown are representatives of three independent experiments. (**F**) Expression of the full length (FL) and fragments of TbTim17 fusion proteins in yeast. Yeast cells co-transformed with a pair of bait and prey plasmids, as indicated, were grown in –Leu/-Trp broth and subjected to protein extraction. Total cellular proteins were analyzed via SDS-PAGE and immunoblot analysis using anti-Myc and anti-HA antibodies to detect BD and AD fusion proteins tagged with the Myc and HA epitopes, respectively. Corresponding Coomassie-stained gels are shown for loading control.

**Figure 5 ijms-25-01415-f005:**
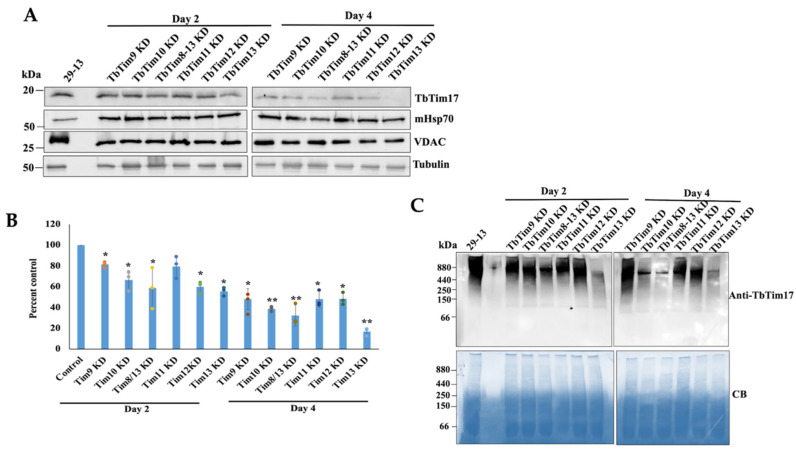
Effect of small TbTims’ knockdown on TbTim17 steady-state levels and TbTIM17 complex integrity. (**A**) The steady-state levels of TbTim17 in the small TbTim RNAi mitochondria. The small TbTim RNAi cells were induced with doxycycline; cells were harvested at day 2 and day 4 after induction for isolation of the mitochondrial fraction. Mitochondrial proteins from the parental (29–13), TbTim9 RNAi (TbTim9 KD), TbTim10 RNAi (TbTim10 KD), TbTim8/13 RNAi (TbTim8/13 KD), TbTim11 RNAi (TbTim11 KD), TbTim12 RNAi (TbTim12 KD), and TbTim13 RNAi (TbTim13 KD) *T. brucei* cell lines were analyzed via immunoblot using TbTim17, Hsp70, VDAC, and β-tubulin antibodies. (**B**) Quantitation of the TbTim17 levels in each RNAi cells line. Densitometric scanning of the TbTim17 and tubulin protein bands was performed using ImageJ software (version 1.53a, National Institute of Health, USA). Band intensities for TbTim17 were normalized with the corresponding tubulin bands and calculated relative to the levels of TbTim17 in the 29–13 mitochondria sample. Values were plotted for each RNAi cell line. Standard errors were calculated from three independent experiments. ** *p* <0.01 and * *p* < 0.05. (**C**) Analysis of the TbTIM17 complex levels in small TbTim knockdown mitochondria. Mitochondria were isolated from the parental (29–13) and the small TbTim RNAi cells grown in the presence of doxycycline for 2 and 4 days. Equal micrograms of mitochondrial proteins were solubilized with digitonin (1%) and analyzed via BN-PAGE followed by immunoblot analysis using anti-TbTim17 antibody. The positions of the molecular weight marker proteins on the gels are shown. Coomassie Blue (CB)-stained gels show equal loading of samples.

**Figure 6 ijms-25-01415-f006:**
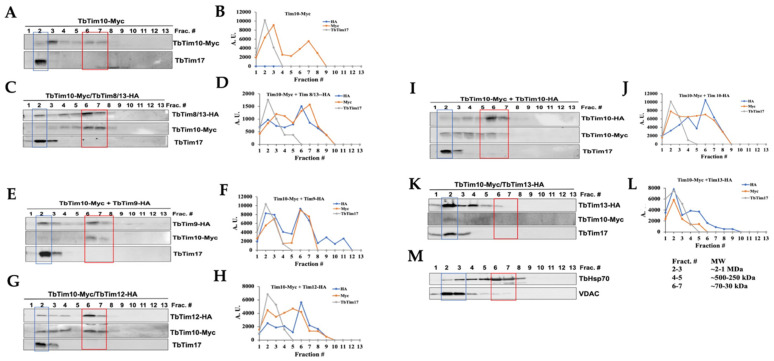
Analysis of the small TbTims complexes in *T. brucei*. Mitochondria isolated from the TbTim10-Myc, TbTim10-Myc/TbTim8/13-HA, TbTim10-Myc/TbTim9-HA, TbTim10-Myc/TbTim12-HA, TbTim10-Myc/TbTim10-HA, and TbTim10-Myc/TbTim13-HA *T. brucei* cells were solubilized with digitonin. The soluble supernatants were fractioned by SEC. Proteins were eluted with 1X Native buffer (20 mM Tris, pH 7.0, 50 mM NaCl, 1 mM EDTA, 10% glycerol, 1 mM PMSF, and 0.1% digitonin). (**A**,**C**,**E**,**G**,**I**,**K**) Fractions (1–13) were analyzed via SDS-PAGE and immunoblot analysis using HA and Myc antibodies. (**M**) mHSP70 and VDAC were used as standard mitochondrial proteins. Fraction numbers (Frac. #) are indicated at the top of the blot. Fractions 2 and 6–7 are in blue and red boxes, respectively. (**B**,**D**,**F**,**H**,**J**,**L**) Myc- and HA-tagged small TbTims and TbTim17 bands were quantitated via ImageJ software (version 1.53a, National Institute of Health, USA) for each run and plotted against the fraction numbers in Excel. Band intensity is presented in arbitrary units (A.U.). Runs from multiple experiments from each sample were reproducible.

**Figure 7 ijms-25-01415-f007:**
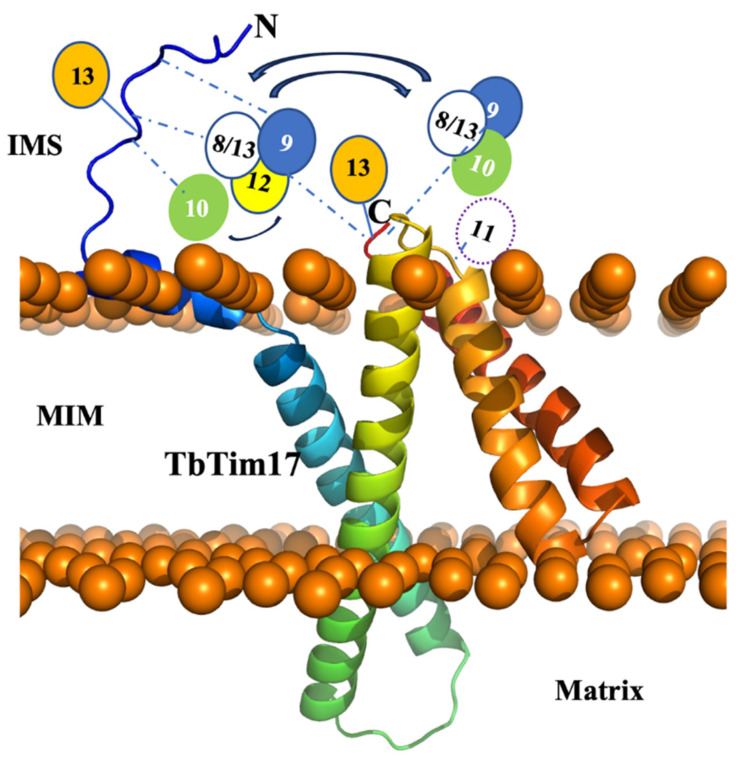
Model for the small TbTims complexes with TbTim17. The TbTim17 model was generated using AlphaFold within the ChimeraX-1.4 software (UCSF ChimeraX, San Francisco, CA, USA). The model was then uploaded to CHARMM-gul server and oriented within a mitochondrial membrane through the OPM server. Using the CHARMM-gul server, the model was assembled and equilibrated. The N- and C-termini are marked. Mitochondrial inner membrane (MIM), intermembrane space (IMS), and matrix are labeled. The putative small TbTim complexes, as, for example, (TbTim9-TbTim8/13-TbTim10) and (TbTim9-TbTim8/13-TbTim12), are placed near the C-terminal of TbTim17 and linked by dashed blue lines, indicating dynamic interactions. Some interactions of the N-terminal domain of TbTim17 with small TbTims are also shown by dashed blue lines. Reversible arrows indicate possible exchange of subunits. TbTim13 stays attached to the N- and C-terminals of TbTim17.

**Table 1 ijms-25-01415-t001:** *T. brucei* small TbTims interactions via Y2H analysis.

	Growth			
Interacting Partners	0 mM AT	2 mM AT	3.5 mM AT	5 mM AT
TbTim9-AD + TbTim9-BD	++	-	-	-
TbTim10-AD + TbTim10-BD	+	-	-	-
TbTim8/13-AD + TbTim8/13-BD	++	++	+	+
TbTim9-AD + TbTim10-BD	++	+	+	-
TbTim9-AD + TbTim8/13-BD	++	++	+	+
TbTim10-AD + TbTim8/13-BD	++	++	+	+
TbTim11-AD-TbTim9-BD	++	++	-	-
TbTim11-AD + TbTim10-BD	++	++	-	-
TbTim11-AD + TbTim11-BD	++	+	-	-
TbTim11-AD + TbTim12-BD	++	++	-	-
TbTim11-AD + TbTim13-BD	++	++	-	-
TbTim8/13-AD + TbTim11-BD	++	++	-	-
TbTim9-AD + TbTim12-BD	++	-	-	-
TbTim10-AD + TbTim12-BD	++	++	-	-
TbTim12-AD + TbTim12-BD	++	+	-	-
TbTim13-AD + TbTim12-BD	++	++	-	-
TbTim8/13-AD + TbTim12-BD	++	++	-	-
TbTim9-AD + TbTim13-BD	++	++	+	+
TbTim10-AD + TbTim13-BD	++	++	-	-
TbTim13-AD + TbTim13-BD	++	+	+	-
TbTim8/13-AD + TbTim13-BD	++	++	-	-

- no growth + low growth ++ high growth.

**Table 2 ijms-25-01415-t002:** *T. brucei* small TbTims’ interactions with different structural domains of TbTim17 via Y2H analysis.

	Growth			
Interacting Partners	0 mM AT	2 mM AT	3.5 mM AT	5 mM AT
TbTim9-AD + TbTim17 N-terminal-BD	++	++	++	+
TbTim10-AD + TbTim17 N-terminal-BD	++	++	+	-
TbTim11-AD + TbTim17 N-terminal-BD	++	++	+	-
TbTim12-AD + TbTim17 N-terminal-BD	++	++	+	-
TbTim13-AD + TbTim17 N-terminal-BD	++	++	+	-
TbTim8/13-AD + TbTim17 N-terminal-BD	++	++	+	-
TbTim9-AD + TbTim17 loop2-BD	-	-	-	-
TbTim10-AD + TbTim17 loop2-BD	-	-	-	-
TbTim11-AD + TbTim17 loop2-BD	+	+	+	+
TbTim12-AD + TbTim17 loop2-BD	-	-	-	-
TbTim13-AD + TbTim17 loop2-BD	-	-	-	-
TbTim8/13-AD + TbTim17 loop2-BD	-	-	-	-
TbTim9-AD + TbTim17 C-terminal-BD	++	++	++	++
TbTim10-AD + TbTim17 C-terminal-BD	++	++	++	++
TbTim11-AD + TbTim17 C-terminal-BD	++	++	++	++
TbTim12-AD + TbTim17 C-terminal-BD	++	++	++	++
TbTim13-AD + TbTim17 C-terminal-BD	++	++	++	++
TbTim8/13-AD + TbTim17 C-terminal-BD	++	++	++	++

- no growth + low growth ++ high growth.

## Data Availability

All data generated from this study are presented in the manuscript and the Appendix A.

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
