# Peer review of "Unique Interactions of the Small Translocases of the Mitochondrial Inner Membrane (Tims) in Trypanosoma brucei"

_ijms, 2024, doi:10.3390/ijms25031415_

Round 1

Reviewer 1 Report

Comments and Suggestions for Authors

The manuscript involves complex work evaluating the interaction between Tims of T. brucei.

It is a well-presented and complete work elegantly demonstrating the established hypothesis.

Suggestion: In the title, replace "Tims" with "translocases fo the inner membrane". It becomes clearer.

Material and Methods: Testration enzyme names should be in italics.

Results; The first time you mention AT (line 133) state which substance AT represents

Author Response

Thanks to the reviewer for carefully reviewing our manuscript. We have revised our manuscript according to your comments/suggestion.

Suggestion: In the title, replace "Tims" with "translocases for the inner membrane". It becomes clearer.

Response: The title of the manuscript has been modified as suggested. The current title is ‘Unique Interactions of the Small Translocases of the Mitochondrial Inner Membrane (Tims) in Trypanosoma brucei’.

Suggestion: Material and Methods: Restriction enzyme names should be in italics.

Response: Done

Comment: Results; The first time you mention AT (line 133) state which substance AT represents

Response: AT represents 3-amino-1,2,4-triazole. We mentioned this in the Result section first time we used.

Reviewer 2 Report

Comments and Suggestions for Authors

The manuscript contains very valuable information regarding the traslocase machinery in T. brucei, the manuscript is well written and present abundant experimental evidence presented in an orderly manner; however the inclusion of the figures as supplementary material makes it difficult to understand immediately and there are small changes and clarification that I would like to see before acceptance

Introduction

Lines 86 - 94

Homology and homologues are constantly mentioned, HMP modeling does not mention the term; apparently in the manuscript it is mentioned as synonymous to identity when referring how much similarity exist between two proteins, I do not not believe homology is the correct term.

Results

The authors chose to present the figures and graphs as supplementary material, that makes the reading difficult to understand unless the reader downloads the figures; I understand  that there are too many figures that would make the manuscript too long and some effort is necessary to make it shorter, but IJMS is a digital journal I do not believe there are space constrains, maybe with the addition of tables and drawings within the main text of the manuscript could make it shorter, an alternative is to make mosaic figures and mosaic graphs included in the manuscript.

Author Response

Thanks to the reviewer for acknowledging that the manuscript is well written and presents abundant experimental evidence in orderly manner.

Comment: The major concern raised by this reviewer is that we put important data as the supplementary material, which makes it difficult to review. 

Response: We have now included most of our data in the main text. The supplementary Figure S1 is now Figure 3A. Supplementary Figure S2 in previous version has been included in the main text as Fig. 6A and B.

Comment: Introduction Lines 86 - 94

Homology and homologues are constantly mentioned, HMP modeling does not mention the term; apparently in the manuscript it is mentioned as synonymous to identity when referring how much similarity exist between two proteins, I do not not believe homology is the correct term.

Response: We agree with the reviewer and modified the term ‘homology’ and ‘homologous’ in the text as follows.

Line 89-93: “Among these, TbTim9, TbTim10, and TbTim8/13 were initially identified from T. brucei genome data base by Hidden Markov prediction tools using yeast and human small Tims as the queries [36]. Tim8/13 contains the features present in both Tim8 and Tim13 in other eukaryotes.”

Line 97-98: “all these small TbTims have similar secondary and tertiary structures with small Tims in yeast and human.”

Comment: Results

The authors chose to present the figures and graphs as supplementary material, that makes the reading difficult to understand unless the reader downloads the figures; I understand  that there are too many figures that would make the manuscript too long and some effort is necessary to make it shorter, but IJMS is a digital journal I do not believe there are space constrains, maybe with the addition of tables and drawings within the main text of the manuscript could make it shorter, an alternative is to make mosaic figures and mosaic graphs included in the manuscript.

Response: Please see our responses to comment 1

Reviewer 3 Report

Comments and Suggestions for Authors

Overall the manuscript by Guillen et al. is very interesting and could help understanding the import of proteins in the parasite T. brucei’s mitochondrion.

The experimental approach is elegant, even if quite outdated. The manuscript is practically totally focused on the interactions of small Tims more than on their function, therefore the title is slightly overrated.
If the authors intended the word “function” as a synonymous of necessary for survival then it could be by and large accepted.
Most of the data presented are sound, however going through the paper has not been easy.

Recovering of the results from the figures was not straightforward.
It was particularly hard to interpret the figures with the text and the legends very far from each other. The choice of the editing with numerous multistrip-multi-panel with almost no repair is particularly bad.

All the panel labels of all the figures, except Fig1, are cut, which did not help.
Figure 2 panels G, H, I in CB gel, please indicate the MW or at least where are 20-25-37 kDa for helping the comparison with the WB.
The full gels (CB stained) are not the same from which the blot have been performed), the Myc and HA WB come from different gels, not from the same one stripped and reblotted, which could be more indicative.
Is Myc Ab less specific than HA Ab or are truncated TIM been produced ? In the full sized WB there are bands which have not been commented in the text and totally omitted in the cut&mounted figure.

Figure 3A: anti-TbPP5 strongly recognizes lower MW bands and there is no mention in the text either.
Figure 3A original (page 6) TbTIM10-myc blot antiHA -> 2 bands are squared, but no bands are inserted in the cut&mounted Figure 3A in the main text. The authors do not explain this.
There is a big discrepancy between the original gels (page 6) and the cut stripes in the mounted figure 3A; at page 7 of the “entire gels”: anti-VADAC and anti-TbPP5 are cut blots not entire WB.

Figure 3B: the full blot at page 8 are more informative than the cut, where the mounted panels are not the same as in the full gels.
The same discordance is found for figure 3B IP-anti Myc between the mounted one and the full gels (page 9), by the way again the anti VDAC are only different cut and not the full length blot.
Slightly better for the lower panel Anti-HA IP, but the full gels of the controls are not presented.
Figure 4F seems coherent with the data in the blot.
Figure 5 A is called Figure 5F in the full length gels, but there is missing the original coomassie stained -> the blot of HSP70 is not clear whether the band is really at that height.

Figure6 -> the originals are only better quality strips than the mounted ones. Fig 6K with the “control” Hsp70 is from which one of the several columns?
Figure 7 is practically useless, since there is no experiment to corroborate the stoichiometry of the complexes.
Supplementary Figure 2 needs to be placed in the main manuscript.
No data on the qRT are presented: which is the reason for the presence of their sequences in supplementary Table 1?
In conclusion, a better presentation of the data after a more careful analysis of the original uncut WB is mandatory to reinforce the major findings and also to really demonstrate the hypothesis on the stoichiometry and strength of binding.

Author Response

The reviewer thinks that the manuscript is interesting, and it will help to understand mitochondrial protein import in T. brucei. Our responses to other comments are given below.

Comment: The experimental approach is elegant, even if quite outdated. The manuscript is practically totally focused on the interactions of small Tims more than on their function, therefore the title is slightly overrated. If the authors intended the word “function” as a synonymous of necessary for survival, then it could be by and large accepted.

Response: We agree with the reviewer. Here we used basic biochemical and molecular biology approaches to understand the composition and association of the small TbTim complexes in T. brucei. Regarding the function, we found that among 6 small TbTims, TbTim13 is most crucial for the stability/biogenesis of the TbTIM17 complex. This is an important function. We are currently working to elucidate the actual mechanism of this process. However, as suggested by the reviewer we removed the word “function” from the title.

Comment: Most of the data presented are sound, however going through the paper has not been easy.

Response: In our revised manuscript, we have included most of the supplemental figures in the manuscript. We also have divided several composite figures in two figures. Hopefully, these modifications will help the flow of the manuscript for readers.

Comment: Recovering of the results from the figures was not straightforward.
It was particularly hard to interpret the figures with the text and the legends very far from each other. The choice of the editing with numerous multistrip-multi-panel with almost no repair is particularly bad.

Response: Sorry for the complexity of the figures. In this manuscript we worked with six small TbTims and TbTim17. Therefore, to determine their interactions, we had to go all possible combinations using different approaches and present our data with muti-panel figures. We think that in the published article each figure will be in the vicinity of the corresponding result section, which will increase the readability of the paper.

As mentioned, we have multiple samples. Therefore, it was not possible to run all samples together in one gel. To have a symmetry in the presentation we had to cut the figures in multiple panels. We always use the size markers to compare one panel from the other and the original uncropped images are provided for review.

Comment: Figure 2 panels G, H, I in CB gel, please indicate the MW or at least where are 20-25-37 kDa for helping the comparison with the WB.
The full gels (CB stained) are not the same from which the blot have been performed), the Myc and HA WB come from different gels, not from the same one stripped and reblotted, which could be more indicative.
Is Myc Ab less specific than HA Ab or are truncated TIM been produced ? In the full sized WB there are bands which have not been commented in the text and totally omitted in the cut & mounted figure.

Response: We have revised Fig. 2G, H, and I by adding MW markers. CB-stained gels are duplicates gels used for the HA-blots. This is a common process used by other investigators to use duplicate gels for such purpose. We often used duplicate blots to probe with anti-Myc and anti-HA antibodies, because the size of the Tim proteins are very similar, any residual bands after de-probe could hamper the actual results. Each experiment have been performed multiple times to confirm our results. We believe that probing duplicate blots didn’t reduce any legitimacy of our results. Anti-Myc polyclonal antibody showed cross-reactions with a few yeast proteins of higher and lower MW (original uncropped images for Fig. 2G, H, I). We used two controls for these blots, Y2H gold transfected with empty pAD and pBD plasmids (lane 1) and Y2H transfected with P53-BD/SV40-T-AD plasmids (lane 2). The specific bands for the small TbTim fusion proteins are present only in the corresponding lanes and not in the controls. We agree that there is a smaller MW band in most of our samples, which appears a breakdown product. We only showed the levels of the full-length fusion proteins with expected MW in our manuscript because only the fusion proteins are supposed to be in the nucleus and are the subject for intermolecular interactions.

Reviewer 4 Report

Comments and Suggestions for Authors

In this manuscript, the architecture of the translocase of the mitochondrial inner membrane translocase from Trypanosoma brucei is studied. However, I found it very difficult to follow the manuscript and there were no strong conclusions about the subunit composition of the complexes. I also think the result section was very difficult to read due to a large amount of experiments combined with a lot of references to other studies.  

Comments on the Quality of English Language

The manuscript needs language editing. Here are some examples but there are many more:

Row 40, 47: inser “the” in front of “mitochondrial protein import machinery”

Row 52 and 96: it should be “the” in front of “TIM23 complex” and “TbTIM17 complex”

Result section, first paragraph: there are different fonts used in the paragraph.

Row 297 “with antibody for ADP/ATP carrier” (change to: an antibody; the ADP/ATP carrier)

Row 298 “levels of AAC mature complex” (do you mean “matured AAC complex?)

Row 304: “TbTim17 and TbTim50 knockdown” (change to: knockdowns)

Row 307: it should be “a non-specific product”

Author Response

Comment: In this manuscript, the architecture of the translocase of the mitochondrial inner membrane translocase from Trypanosoma brucei is studied. However, I found it very difficult to follow the manuscript and there were no strong conclusions about the subunit composition of the complexes. I also think the result section was very difficult to read due to a large amount of experiments combined with a lot of references to other studies. 

Response: This is an extension of our previous studies on TbTim9, TbTim10, and TbTim8/13. Therefore, we had to mention our published paper and a related paper authored by different investigators in several places in the manuscript. We believe that we made a significant contribution in this manuscript towards the understanding of the small TbTim complexes and their association with TbTim17. 1) We showed that TbTim13 is associated with the larger TbTIM17 complex and essential for its integrity, 2) other small TbTims form smaller 70 kDa complexes and capable to exchange subunits among themselves, 3) The smaller 70 kDa complex can be separated from the larger complex, and 4) each of these small TbTims co-precipitate a part of TbTim17, suggesting that unlike TbTim13 other small TbTims dynamically associates with the larger TbTIM17 complex. Together, we created a structural model that shows for the first time how small TbTims interact and associate with TbTim17.

All language corrections are incorporated.

Reviewer 5 Report

Comments and Suggestions for Authors

I must admit that I liked this manuscript. It is a good example of a biochemical study well-done. I have a few rather cosmetic comments that are aimed at improving the presentation. 

1) I do understand that it is T. brucei-focused. But what is known about small Tims in other species of Trypanosomatidae? How conserved they are? 

2) I found strange that section Results contains parts that clearly belong to Methods. F.e. lns. 117-131 and elsewhere. 

3) Authors may need to justify using Y2H as a chosen method (yeasts and trypanosomes are fundamentally different). In this sense, co-IP experiments are more trustworthy, I think. 

4) Some references are wrongly or inconsistently formatted (e.g. 9, 14, 33, and many others; make sure species and generic names are Italicized, word capitalization, etc.)

5) Labeling of panels in figures is disastrous. They must be uniform (font, size, etc.)    

Author Response

 This reviewer liked the biochemical approaches taken in this manuscript to analyze the interactions of the small Tims in T. brucei. Our responses to the reviewer’s other comments are given below.

Comment: I do understand that it is T. brucei-focused. But what is known about small Tims in other species of Trypanosomatidae? How conserved they are? 

Response: Small Tim family proteins are present in other species of trypanosomatids. However, besides Tim1 in Leishmania, there are not many studies. We have mentioned this in the in the introduction (line 98-99) in our revised manuscript.

Comment: I found strange that section Results contains parts that clearly belong to Methods. F.e. lns. 117-131 and elsewhere. 

Response: In this paragraph, we added a short description of the experiment to explain how we differentiate stronger/weaker intermolecular interactions. Other details are given in the materials and methods.

Comment: Authors may need to justify using Y2H as a chosen method (yeasts and trypanosomes are fundamentally different). In this sense, co-IP experiments are more trustworthy, I think. 

Response: That is right. For this reason, we verified our Y2H results using co-immunoprecipitation analysis.

Comment: Some references are wrongly or inconsistently formatted (e.g. 9, 14, 33, and many others; make sure species and generic names are Italicized, word capitalization, etc.)

Response: We have corrected the format of the references.

Comment: Labeling of panels in figures is disastrous. They must be uniform (font, size, etc.)    

Response: Corrected.

Round 2

Reviewer 3 Report

Comments and Suggestions for Authors

The authors have made their best to improve the readability of the manuscript, however IJMS manuscript instructions have not been followed. Moreover a careful check of the uploaded PDF should have been performed, since the figures are completely messed up and again they have been placed at the very end of the manuscript. This is index of being in a rush for publishing and a symptom of not caring about the reviewers.

According to all the tables and yeast growth figures, the summary scheme in the last figure should add TbTim11 in the 70 kDa complex.

Moreover, even if the association with the large TbTim17 complex is mostly evident with the C-terminal arm, there are also some some minor differences, which have not been pointed our in the text nor in the scheme. For example, TbTim11 is the only one to interact with the loop region of TbTim17 (even if very faintly), which could act as a platform for specific recognition. A similar situation could be seen for the N-terminal stretch which is a hub for TbTim9, -12 and -13. I understand the possible folding problems with TbTim17 Full Length, but TbTim10, -8/13 and -12 are growing, even if very faintly. These data are consistent with the RNAi results, but are not discussed nor wrapped up in the conclusions.

The legends should be more explicit in describing the multi-panel figures, given that many panels are subdivided in small pieces due to the experimental setup, therefore a better self-explanation is needed.

A legend is also needed for each of the full gels or at least a reference to the text paragraph or figure panel/subpanel. There are still many lanes with bands and without labeling. What is the message a reader should understand?

Comments on the Quality of English Language

Please read carefully your manuscript after having included all the revisions (i.e. the clean version), since in many places either the subject or the verb are missing.

Author Response

Responses to the reviewer 4 (2nd review)

Comments: The authors have made their best to improve the readability of the manuscript, however IJMS manuscript instructions have not been followed. Moreover a careful check of the uploaded PDF should have been performed, since the figures are completely messed up and again they have been placed at the very end of the manuscript. This is index of being in a rush for publishing and a symptom of not caring about the reviewers.

Response: Thanks to acknowledge our efforts to increase the readability of our manuscript. The revised manuscript is now placed on IJMS template. Therefore, Figures are now closer to the result section. Sorry, we missed this guideline. We thought the formatting will be done by the journal office once the paper has been accepted. We see the figures clear and organized. 

Comment: According to all the tables and yeast growth figures, the summary scheme in the last figure should add TbTim11 in the 70 kDa complex.

Response: We assess the presence of small Tims in 70 kDa complex by size exclusion chromatography. Since, TbTim11 was not included for SEC analysis due to unavailability of the specific clone, we couldn’t include TbTim11 in the 70 kDa complex.

Comment: Moreover, even if the association with the large TbTim17 complex is mostly evident with the C-terminal arm, there are also some some minor differences, which have not been pointed our in the text nor in the scheme. For example, TbTim11 is the only one to interact with the loop region of TbTim17 (even if very faintly), which could act as a platform for specific recognition. A similar situation could be seen for the N-terminal stretch which is a hub for TbTim9, -12 and -13. I understand the possible folding problems with TbTim17 Full Length, but TbTim10, -8/13 and -12 are growing, even if very faintly. These data are consistent with the RNAi results but are not discussed nor wrapped up in the conclusions.

Response: We have modified our working model as suggested by the reviewer. In our previous version of the manuscript, we did not include the following interactions 1) TbTim11 with loop2 and 2) TbTim9, -12, and -13 with the N-terminal of TbTim17, because these are shown only by Y2H analysis. However, we agree with the reviewer’s comments and included these interactions in our revised manuscript (Fig. 9), where we showed TbTim11 with dotted circle, to indicate weaker interactions. We have included additional discussion regarding these weak interactions (lines 778-779 and 825-829).

Comment: The legends should be more explicit in describing the multi-panel figures, given that many panels are subdivided in small pieces due to the experimental setup, therefore a better self-explanation is needed.

Response: We have included our explanation for creating multiple panels in the figure legends.

Comment: A legend is also needed for each of the full gels or at least a reference to the text paragraph or figure panel/subpanel. There are still many lanes with bands and without labeling. What is the message a reader should understand?

Response: Since the uncropped images are not for publications, we just boxed and labelled the regions that were cropped. We also added the corresponding figure title for the uncropped full gel pictures. We indicated the non-specific bands in the figure legends (Fig. 2).

Comment: Please read carefully your manuscript after having included all the revisions (i.e. the clean version), since in many places either the subject or the verb are missing.

Response: We have checked the grammar again.